# A modular approach to integrating multiple data sources into real-time clinical prediction for pediatric diarrhea

Ben J Brintz[1,2]\*, Benjamin Haaland[3], Joel Howard[4], Dennis L Chao[5], Joshua L Proctor[5], Ashraful I Khan[6], Sharia M Ahmed[2], Lindsay T Keegan[1], Tom Greene[1], Adama Mamby Keita[7], Karen L Kotloff[8], James A Platts-Mills[9], Eric J Nelson[10,11], Adam C Levine[12], Andrew T Pavia[4], Daniel T Leung[2,13]\*

[1]Division of Epidemiology, Department of Internal Medicine, University of Utah, Salt Lake City, United States; [2]Division of Infectious Diseases, Department of Internal Medicine, University of Utah, Salt Lake City, United States; [3]Population Health Sciences, University of Utah, Salt Lake City, United States; [4]Division of Pediatric Infectious Diseases, University of Utah, Salt Lake City, United States; [5]Institute of Disease Modeling, Bill and Melinda Gates Foundation, Seattle, United States; [6]International Centre for Diarrhoeal Disease Research, Bangladesh, Dhaka, Bangladesh; [7]Centre Pour le Développement des Vaccins-Mali, Bamako, Mali; [8]Division of Infectious Disease and Tropical Pediatrics, University of Maryland, Baltimore, United States; [9]Division of Infectious Diseases and International Health, University of Virginia, Charlottesville, United States; [10]Departments of Pediatrics, University of Florida, Gainesville, United States; [11]Departments of Environmental and Global Health, University of Florida, Gainesville, United States; [12]Department of Emergency Medicine, Brown University, Providence, United States; [13]Division of Microbiology and Immunology, Department of Internal Medicine, University of Utah, Salt Lake City, United States

\*For correspondence:
ben.brintz@hsc.utah.edu (BJB);
daniel.leung@utah.edu (DTL)

**Competing interests:** The authors declare that no competing interests exist.

**Abstract** Traditional clinical prediction models focus on parameters of the individual patient. For infectious diseases, sources external to the patient, including characteristics of prior patients and seasonal factors, may improve predictive performance. We describe the development of a predictive model that integrates multiple sources of data in a principled statistical framework using a post-test odds formulation. Our method enables electronic real-time updating and flexibility, such that components can be included or excluded according to data availability. We apply this method to the prediction of etiology of pediatric diarrhea, where 'pre-test' epidemiologic data may be highly informative. Diarrhea has a high burden in low-resource settings, and antibiotics are often over-prescribed. We demonstrate that our integrative method outperforms traditional prediction in accurately identifying cases with a viral etiology, and show that its clinical application, especially when used with an additional diagnostic test, could result in a 61% reduction in inappropriately prescribed antibiotics.

## Introduction

Healthcare providers use clinical decision support tools to assist with patient diagnosis, often to improve accuracy of diagnosis, reduce cost by avoiding unnecessary laboratory tests, and in the case of infectious diseases, deter the inappropriate prescription of antibiotics (*Sintchenko et al., 2008*). Typically, data entered into these tools is related directly to the patient's individual characteristics,

but data sources external to the patient can be informative for diagnosis. For example, climate, seasonality, and epidemiological data inform predictive models for communicable disease incidence (*Colwell, 1996*, *Chao et al., 2019 Fine et al., 2011*). The emergence of advanced computing and machine learning has enabled the incorporation of large data sources in the development of clinical support tools (*Shortliffe and Sepúlveda, 2018*) such as SMART-COP for predicting the need for intensive respiratory support for pneumonia (*Charles et al., 2008*) or the ALaRMS model for predicting inpatient mortality (*Tabak et al., 2014*).

Clinical decision support tools rely on the availability of information sources and computing at the time of patient encounter. Although increased availability of internet/mobile phones have increased access to information and computing power in low-resource settings, there may be times when connectivity, computing power, or data-collection infrastructure is unavailable. Thus, there is a need to build clinical decision support tools which can flexibly include features of external sources when available, or function without them if unavailable. Methods that enable the dynamic updating of predictive models are advantageous due to potential cyclical patterns of infectious etiologies. Furthermore, with the emergence of point-of-care (POC) tests for clinical decision-making (*Price, 2001*), predictive models that are able to integrate results of such diagnostic testing could enhance their usefulness.

We develop a novel method for diagnostic prediction which integrates multiple data sources by utilizing a post-test odds formulation with proof-of-concept in antibiotic stewardship for pediatric diarrhea. Our formulation first fits separate models from different sources of data, and then combines the likelihood ratios from each of these independent models into a single prediction. This method allows the multiple components to be flexibly included or excluded. We apply this method to the prediction of diarrhea etiology with data from the Global Enteric Multicenter Study (GEMS) (*Kotloff et al., 2013*) and assess the performance of this tool, including with the addition of a synthetic diagnostic, using two forms of internal-validation and by showing its potential effect on reducing inappropriate antibiotic use.

## Materials and methods

We present our approach to building and assessing a flexible multi-source clinical prediction tool with (1) the data sources, (2) the individual prediction models, (3) the use of the likelihood ratio for integrating predictive models, (4) validation of the method, (5) the impact of an additional diagnostic, and (6) a simulation of conditionally dependent tests. We program our prediction tool using R version 3.6.2 (R Project for Statistical Computing, RRID:SCR_001905).

### Data sources

We apply our post-test odds model using clinical data from GEMS, a prospective, case-control study from 2007 to 2011 which took place in seven countries in Africa and Asia. Methods for the GEMS study have been described in detail (*Kotloff et al., 2012*). Briefly, 9439 children with moderate-to-severe diarrhea were enrolled at local health care centers along with one to three matched control-children. A fecal sample was taken from each child at enrollment to identify enteropathogens and clinical information was collected, including demographic, anthropometric, and clinical history of the child. We used the quantitative real-time PCR-based (qPCR) attribution models developed by *Liu et al., 2016* in order to best characterize the cause of diarrhea. Our dependent variable was presence or absence of viral etiology, defined as a diarrhea episode with at least one viral pathogen with an episode-specific attributable fraction (AFe $\geq$ 0.5) and no bacterial or parasitic pathogens with an episode-specific attributable fraction. Prediction of viral attribution is clinically meaningful since it indicates that a patient would not benefit from antimicrobial therapy. We defined other known etiologies as having a majority attribution of diarrhea episode by at least one other non-viral pathogen. We exclude patients with unknown etiologies when fitting the model, though it has been previously shown that these cases have a similar distribution of viral predictions using a model with presenting patient information as those cases with known etiologies (*Brintz et al., 2020*).

We obtained weather data local to each site's health centers during the GEMS study using NOAA's Integrated Surface Database (*Smith et al., 2011*). The incidence of many pathogens, including rotavirus (*Cook et al., 1990*), norovirus (*Ahmed et al., 2013*), cholera (*Emch et al., 2008*), and *Salmonella* (*Mohanty et al., 2006*), are known to have seasonal patterns, and other analyses have

established climatic factors to be associated with diarrheal diseases (*Colwell, 1996*, *Chao et al., 2019*, *Farrar et al., 2019*). Stations near GEMS sites such as in The Gambia exhibit seasonal patterns (*Figure 1*). We used daily temperature and rain data weighted most by those weather stations closest to the GEMS sites (Appendix 1).

## Construction of predictive models

We define each model using the features described in the below sub-sections in an additive logistic regression model. Each model can be trained using a sample of data from a specific country, continent, or all available data.

### Predictive model (A) presenting patient

The patient model derived from the GEMS data treats each enrolled patient as an observation and uses their available patient data at presentation to predict viral only versus other etiology of their infectious diarrhea. In order to make a parsimonious model, we used the previously published random forests variable importance screening (*Brintz et al., 2020*). Using the screened variables (*Table 1*), we fit a logistic regression including the top five variables that would be accessible to providers at the time of presentation. These include age, blood in stool, vomiting, breastfeeding status, and mid-upper arm circumference (MUAC), an indicator of nutritional status. We note that while variables such as fever and diarrhea duration were shown to be important in previous studies (*Fontana et al., 1987*), adding these variables did not improve performance. Additionally, we excluded 'Season', since variables representing it are included in the climate predictive model (discussed below), as well as 'Height-for-age Z-score', another indicator of nutritional status, which would require a less feasible calculation than measurement of MUAC.

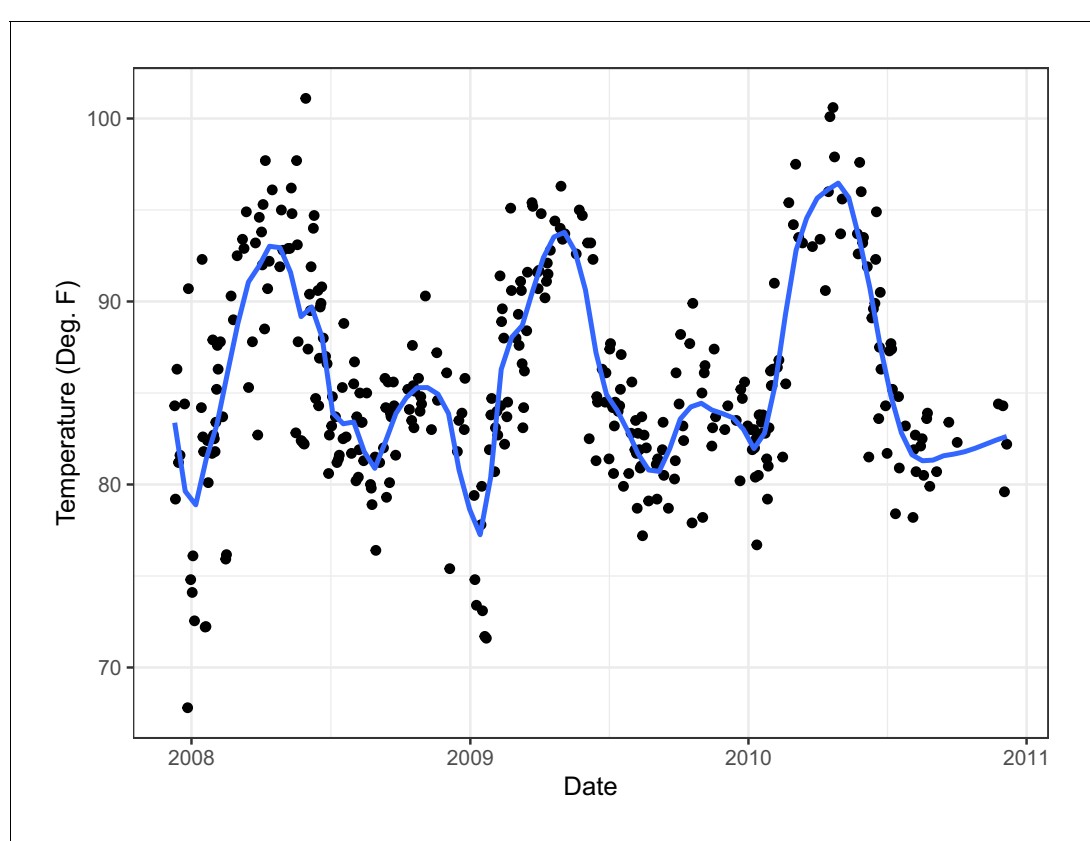

**Figure 1.** Temperature in The Gambia over study period with (blue) trend line from LOESS (locally estimated scatterplot smoothing).
The online version of this article includes the following figure supplement(s) for figure 1:

**Figure supplement 1.** The black line represents a 2-week rolling average of daily viral etiology rates over time.
**Figure supplement 2.** The black line represents a 2-week rolling average of daily viral etiology rates over time.

**Table 1.** Rank of variable importance by average reduction in the mean squared prediction error of the response using Random Forest regression.

Greyed rows are variables that would be accessible for providers in LMICs at the time of presentation. *Table 1* is reproduced from *Brintz et al., 2020*, PLoS Negl Trop Dis., published under the Creative Commons Attribution 4.0 International Public License (CC BY 4.0; https://creativecommons.org/licenses/by/4.0/).

**Viral etiology**

| Variable name | Variance reduction |
| --- | --- |
| Age | 51.6 |
| Season | 29.0 |
| Blood in stool | 26.1 |
| Height-for-age Z-score | 24.7 |
| Vomiting | 23.0 |
| Breastfeeding | 22.0 |
| Mid-upper arm circumference | 20.9 |
| Respiratory rate | 18.5 |
| Wealth index | 18.3 |
| Body Temperature | 16.7 |

## Predictive model (B) climate

We use an aggregate (mean) of the weighted (Appendix 1) local weather data over the prior 14 days to create features that capture site-specific climatic drivers of etiology of infectious diarrhea. By taking an aggregate, we create a moving average that reflects the seasonality seen in *Figure 1*. An example of the aggregate climate data from The Gambia is shown in *Figure 1—figure supplement 1*. From the figure, which also shows a moving average of the viral rate, We see that the periods of higher viral cases of diarrhea tend to have low temperatures and less rain.

## Predictive model (C) seasonality

We include a predictive model with sine and cosine functions as features as explored in *Stolwijk et al., 1999*. Assuming a periodicity of 365.25 days, we have functions $sin(\frac{2\pi t}{365.25})$ and $cos(\frac{2\pi t}{365.25})$. We show that standardized seasonal sine and cosine curves correlate with a rolling average of daily viral etiology rates in The Gambia over time (*Figure 1—figure supplement 2*). These functions can be used to model the country-specific seasonality of viral etiology rate.represent multiple underlying processes that result in a seasonality of viral etiology.

## Use of the likelihood ratio to integrate predictive models from multiple data sources

We integrate predictive models from the multiple sources of data described above using the post-test odds formulation. Using Bayes' Theorem, $P(A|B) = \frac{P(B|A) \cdot P(A)}{P(B)}$, to construct the post-test odds of having a viral etiology,

$$\frac{P(V=1|T_1=t_1, T_2=t_2, \cdots, T_k=t_k)}{P(V=0|T_1=t_1, T_2=t_2, \cdots, T_k=t_k)} = \frac{P(V=1, T_1=t_1, T_2=t_2, \cdots, T_k=t_k)}{P(V=0, T_1=t_1, T_2=t_2, \cdots, T_k=t_k)} \tag{1}$$

$$= \frac{P(T_1=t_1, T_2=t_2, \cdots, T_k=t_k|V=1) \cdot P(V=1)}{P(T_1=t_1, T_2=t_2, \cdots, T_k=t_k|V=0) \cdot P(V=0)} \tag{2}$$

$$= \frac{P(V=1)}{P(V=0)} \cdot \prod_{j=1}^{k} \frac{P(T_j=t_j|V=1)}{P(T_j=t_j|V=0)} \tag{3}$$

where $V=1$ represents a viral etiology and $V=0$ represents an other known etiology, $T_1, T_2, \cdots, T_k$ represent the $k$ tests, the distribution of the predictions from one or more predictive models, used

to obtain the post-test odds, and $\frac{P(V=1)}{P(V=0)}$ is the pre-test odds. Note that going from line (2) to line (3) requires conditional independence between the tests, that is, that $P(T_i = t_i, T_j = t_j | V = 1) = P(T_i = t_i | V = 1) \cdot P(T_j = t_j | V = 1)$ and $P(T_i = t_i, T_j = t_j | V = 0) = P(T_i = t_i | V = 0) \cdot P(T_j = t_j | V = 0)$ for all $i$ and $j$. We test for conditional independence to assess the necessity of making higher-dimensional kernel density estimates using the *ci.test* function from the {*bnlearn*} package in R (**Scutari, 2010**). We derive each $P(T_j = t_j | V = 1)$ and $P(T_j = t_j | V = 0)$ using Gaussian kernel density estimates on conditional predictions from a logistic regression model fit on the training set (**Silverman, 1986**). The distribution of $P(T_j | V)$ is derived using the kernel density estimator $f(t_j) = \frac{1}{nh} \sum_{i=1}^{n} K\left(\frac{t_j - x_i}{h}\right)$ where, in our case, $K(x) = \phi(x)$, the standard normal density function, and the bandwidth, $h$, is Silverman's 'rule of thumb' and the default chosen in the *density* function in R (**Parzen, 1962**).

*Figure 2* shows an example of the frequency of predictions from a logistic regression model conditional on the viral-only status ($V = 0$ and $V = 1$) determined from attributable fractions. Additionally, we overlaid the estimated one-dimensional kernel density. To obtain the value of $\frac{P(T_j = t_j | V = 1)}{P(T_j = t_j | V = 0)}$, the predicted odds, from a model's prediction, we divide the kernel density estimate from the $V = 1$ set (right) by the kernel density estimate from the $V = 0$ set (left). It is feasible to estimate a multi-dimensional kernel density so that it is not necessary to make the conditional independence assumption to move from line 2 to line 3 in the equation above. *Figure 2—figure supplement 1* shows an example two-dimensional contour plot for kernel density estimates of predicted values obtained from logistic regression on GEMS seasonality and climate data in Mali which we will discuss further below. The density was created using R function *kde2d* (**Venables and Ripley, 2002**).

## Pre-test odds from historical data

We calculated pre-test odds using historical rates of viral diarrhea by site and date. We utilize available diarrhea etiology data for a given date, regardless of year, and site using a moving average such that pre-test probability $\pi_d$ for date $d$ is

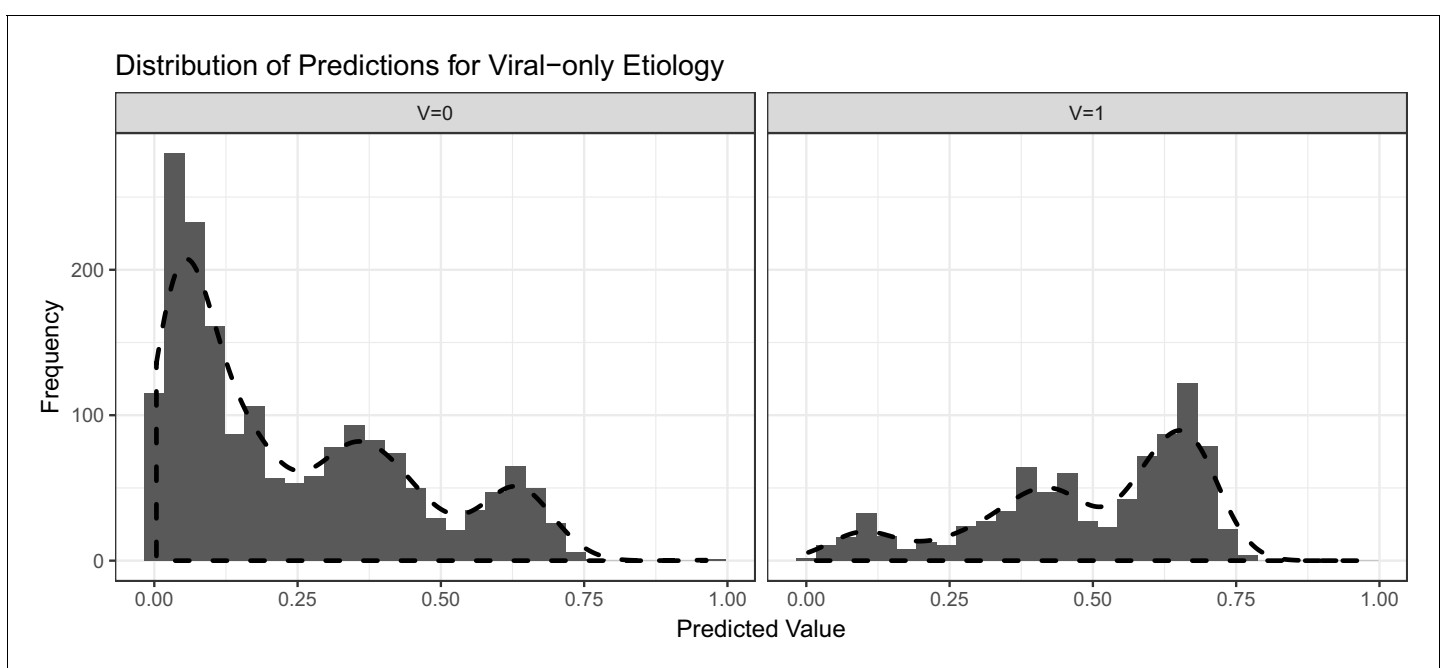

**Figure 2.** Histograms with overlaid estimated kernel densities (dashed lines) of predicted values obtained from logistic regression on patient training data. The left graph represent other known etiologies and the right graph represent viral etiologies. The dashed lines do not represent standardized density heights so the heights for V = 0 and V = 1 should not be compared from this graph.

The online version of this article includes the following figure supplement(s) for figure 2:

**Figure supplement 1.** Contour plots of two-dimensional kernel densities of predicted values obtained from logistic regression on GEMS climate and seasonality data in Mali.

$$\pi_d = \frac{D_{d-n} + D_{d-n+1} + \cdots + D_d + \cdots + D_{d+n-1} + D_{d+n}}{k_{d-n} + k_{d-n+1} + \cdots + k_d + \cdots + k_{d+n-1} + k_{d+n}}$$
$$D_d = \Sigma_{i=1}^{k_d} D_{di}$$

where $k_d$ is the number of observed patients on date $d$, $D_{di}$ is 1 if the etiology of the patients' diarrhea is viral and 0 otherwise, and $n$ is the number of days included on both sides of the moving average. We would expect $\pi_d$ to represent a pre-test probability of observing a viral diarrhea etiology on date $d$. Given that this rate information will likely be unavailable in new sites without established etiology studies, we provide an alternative formula based on recent patients' presentations (Appendix 2). Additionally, we include a sensitivity analysis by calculating pre-test odds using conventional diagnostic methods data as qPCR data are unlikely to be available in high-burden settings.

## Validating the method

Given the temporal nature of some of the tests we developed, we estimate model performance using within rolling-origin-recalibration evaluation. This method evaluates a model by sequentially moving values from a test set to a training set and re-training the model on all of the training set (*Bergmeir and Benítez, 2012*); for example, we train on the first 70% of the data and test on the remaining 30%, then train on the first 80% of the data and test on the remaining 20%. No data from the training set is used as part of the prediction for the test set. In each iteration of evaluation, predictions on the test set are produced and corresponding measures of performance obtained: the receiver operating characteristic (ROC) curve, and area under the ROC curve (AUC), also known as the C-statistic, along with AUC confidence intervals (*LeDell et al., 2015*). *Figure 3* depicts one iteration of within rolling-origin-recalibration evaluation.

We additionally include a joint density for the climate and seasonal data in which we estimate a two-dimensional kernel density (not shown in *Figure 3*). This model is called "Joint' in the results to follow. To assess how this model might generalize to a site that was not used for model training, we used a leave-one-site-out validation. By excluding a site and training the model's tests at a higher level, such as on the entire continent, we get an idea of performance at a new site within one of the continents for which we have data. Lastly, we define a threshold for the predicted odds ratio based on the desired specificity of the model. We use this threshold to evaluate the effect of the model on prescription or treatment of patients with antibiotics in the GEMS data.

## Modeling the impact of an additional diagnostic test

We include a theoretical diagnostic which indicates viral versus other etiology with a given sensitivity and specificity specifically to show the effect of an additional diagnostic-type test, such as a host biomarker-based point-of-care stool testpoint-of-care stool test, on the performance of our integrated post-test odds model. We include three scenarios: (1) 70% sensitivity and 95% specificity, (2) 90% sensitivity and 95% specificity, and (3) 70% sensitivity and 70% specificity. In order to estimate the performance of an additional diagnostic test, for each patient in each of 500 bootstrapped samples of our test data, we randomly simulated a test result based on the sensitivity or specificity of the diagnostic test. From the simulated test result, we derive the likelihood ratio of the component directly from the specified sensitivity and specificity of the test. A positive test results in a component likelihood ratio of $\frac{sensitivity}{1-specificity}$ and a negative test results in a component likelihood ratio of $\frac{1-sensitivy}{specificity}$. We then take an average the measure of performance of the bootstrapped samples.

## Simulation of conditionally dependent tests

We demonstrate the utility of the two-dimensional kernel density estimate through simulation. In each iteration of the simulation (100 iterations), we generate 3366 responses from a random Bernoulli variable $Z$ with a $\frac{1}{3}$ probability of success (the approximate proportion of GEMS cases with a viral etiology). Then, conditioned on Z we generate predictive variables X and Y such that:

$$X = Z + \sigma \tag{4}$$
$$Y = \gamma \dot{X} + Z + \sigma \tag{5}$$

where $\sigma$ is a random draw from the standard normal distribution and values of $\gamma$ ranging from $-10$

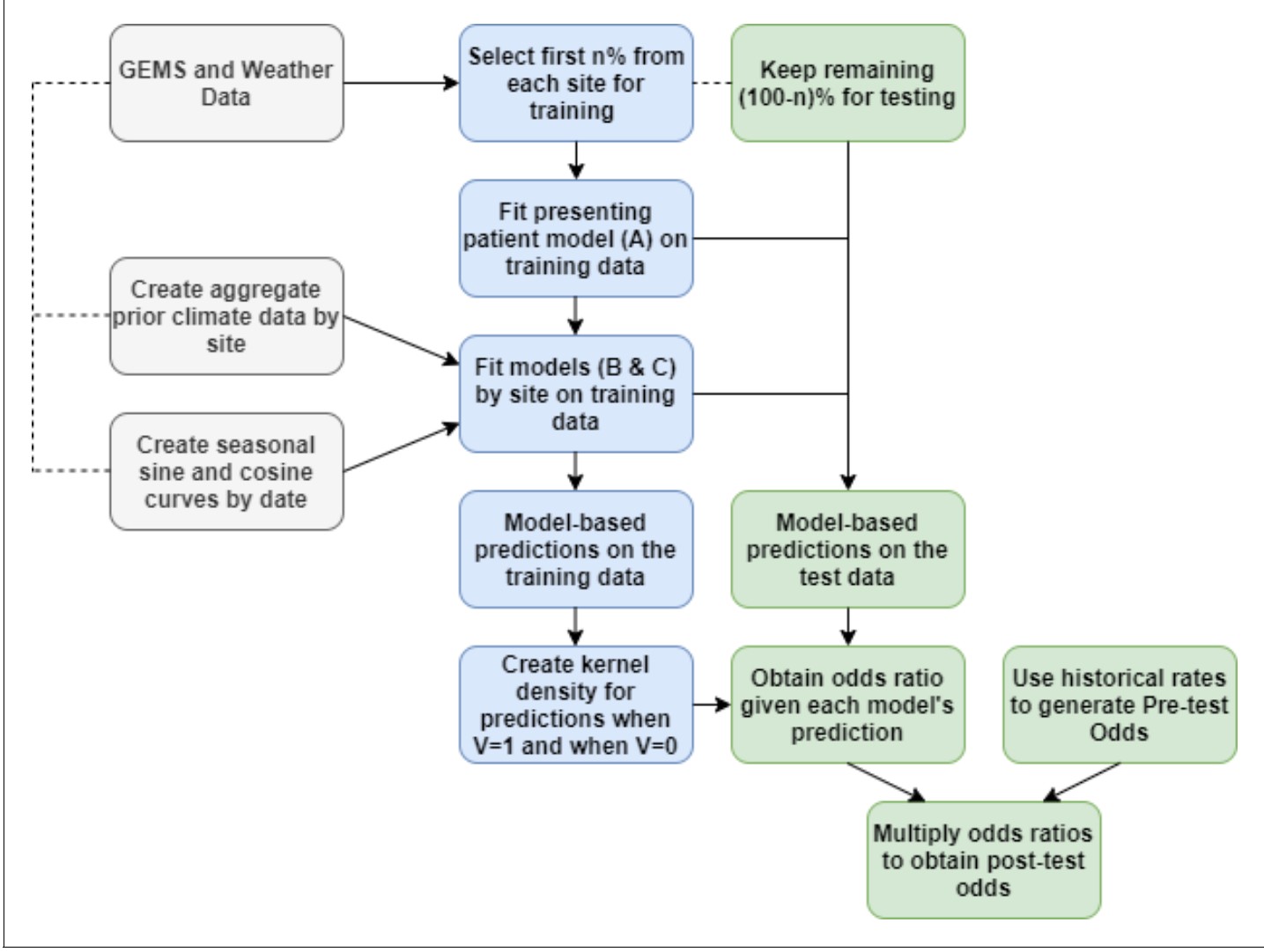

**Figure 3.** The steps for fitting prediction models and calculating the post-test odds for within rolling-origin-recalibration evaluation.

to 10 determine the level of conditional dependence between the two predictors conditional on the value of Z. $\gamma = 0$ indicates conditional independence. Using an 80% training set, we derive the kernel density estimate for the likelihood ratio (no pre-test odds included) using X and Y as two separate tests and as a single two-dimensional test and calculate the AUC from the 20% test set.

## Determination of appropriate antibiotic prescription

We demonstrate the clinical usefulness of our models by applying them directly to the prescription of antibiotics. For each version of the model, we determined the threshold of prediction that would amount to attaining a model specificity of 0.90 and 0.95. Since the prediction of a viral only etiology of diarrhea indicates that antibiotics should not be prescribed, we chose these high specificities due to the potential harm or even death that could occur if a patient who needed antibiotics did not receive them. Using the thresholds, we determine which patients our models would correctly predict a viral only etiology of their diarrhea (true positives) as well as patients our model would incorrectly predict a viral only etiology of their diarrhea (false positives).

## Results

### Integrative post-test odds models outperformed traditional models for prediction of diarrhea etiology

Of the 3366 patients in GEMS with an attributable identified pathogen, 1049 cases were attributable to viral only etiology. We first examined whether our integrative post-test odds model can better discriminate between patients with diarrhea of viral-only etiology and patients with other etiologies than a traditional prediction model which includes only the presenting patient's information. We found that the best integrative model with an AUC of 0.839 (0.808–0.870) had a statistically better performance than the traditional model with an AUC of 0.809 (0.776–0.842) with a p-value of 0.01 (DeLong, two-sided). Overall, using the AUC as a discrimination metric, the integrative models (AUC: 0.837 (0.806–0.869)) outperformed the traditional model (AUC: 0.809 (0.776–0.842)). Overall, the best performing models were ones in which either the seasonal sine and cosine curves, or the prior patient pre-test component alone was added to the presenting patient information with AUC's of 0.830 and 0.839 (with 80% training data), respectively (*Figure 4*). Including additional components, especially including both climate and seasonality (although not as a joint density), appears to reduce the performance. As expected, a reduced testing set increases the AUC but also increases the variance of the estimate (*Figure 4—figure supplement 1*). Using conventional diagnostic methods data data to calculate pre-test odds instead of qPCR data reduces AUC slightly from 0.839 to 0.829 (0.798–0.860).

To assess our model's performance more granularly, we then examine performance of the top two predictive models by individual sites. We found that the AUC, with 80% training and 20% testing, varied greatly by site, ranging from 0.63 in Kenya to 0.95 in Bangladesh (*Table 2*). Of note, the

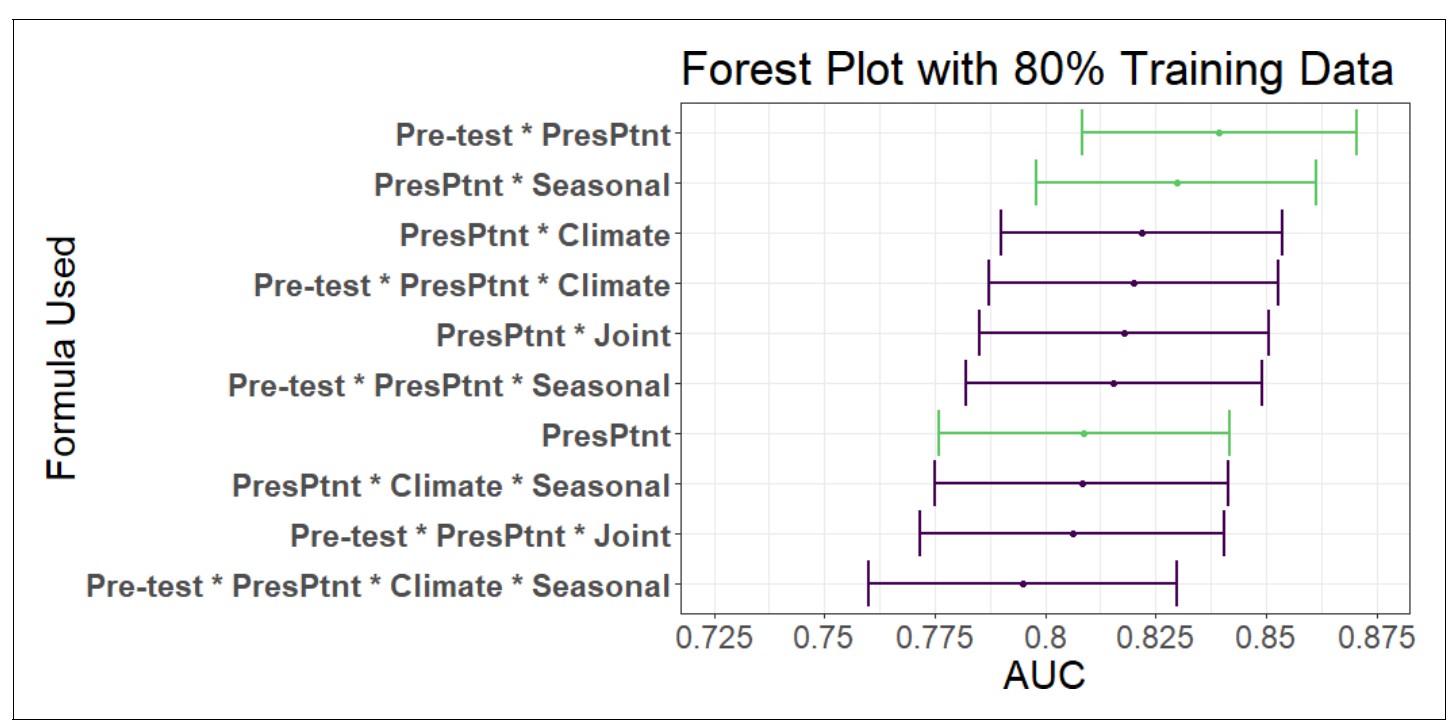

**Figure 4.** AUC's and confidence intervals for post-test odds used in the 80% training and 20% testing iteration. 'PresPtnt' refers to the predictive model using the presenting patient's information. 'Pre-test' refers tot he use of pre-test odds based on prior patients' predictive models. 'Climate' refers to the predictive model using aggregate local weather data. 'Seasonal' refers to the predictive model based on seasonal sine and cosine curves. 'Joint' refers to the two-dimensional kernel density estimate from the Seasonal and Climate predictive models.

The online version of this article includes the following source data and figure supplement(s) for figure 4:

**Source data 1.** AUC's and confidence intervals for post-test odds used in the 80%training and 20%testing iteration.

**Figure supplement 1.** AUC's and confidence intervals for tests used in within rolling-origin-recalibration evaluation.

**Figure supplement 2.** AUC's and confidence intervals for tests used in the leave-one-site-out evaluation.

**Table 2.** AUC results by site using 80% of data for training and 20% of data for testing of the top two models.

PresPtnt refers to the model fit using presenting patient information.

| Country | Test set size | Formula | AUC (95% CI) |
|---|---|---|---|
| Kenya | 79 | Pre-test * PresPtnt | 0.65 (0.53–0.77) |
| | | PresPtnt * Seasonal | 0.66 (0.54–0.78) |
| | | PresPtnt | 0.63 (0.51–0.75) |
| Mali | 88 | Pre-test * PresPtnt | 0.74 (0.61–0.86) |
| | | PresPtnt * Seasonal | 0.78 (0.66–0.89) |
| | | PresPtnt | 0.75 (0.62–0.87) |
| Pakistan | 108 | Pre-test * PresPtnt | 0.81 (0.72–0.89) |
| | | PresPtnt * Seasonal | 0.8 (0.72–0.88) |
| | | PresPtnt | 0.81 (0.73–0.89) |
| India | 119 | Pre-test * PresPtnt | 0.84 (0.76–0.91) |
| | | PresPtnt * Seasonal | 0.85 (0.78–0.92) |
| | | PresPtnt | 0.81 (0.74–0.89) |
| The Gambia | 80 | Pre-test * PresPtnt | 0.89 (0.82–0.96) |
| | | PresPtnt * Seasonal | 0.87 (0.79–0.94) |
| | | PresPtnt | 0.78 (0.67–0.88) |
| Mozambique | 66 | Pre-test * PresPtnt | 0.88 (0.79–0.97) |
| | | PresPtnt * Seasonal | 0.9 (0.82–0.98) |
| | | PresPtnt | 0.77 (0.66–0.89) |
| Bangladesh | 141 | Pre-test * PresPtnt | 0.91 (0.82–1) |
| | | PresPtnt * Seasonal | 0.93 (0.88–0.99) |
| | | PresPtnt | 0.95 (0.92–0.99) |

African sites have fewer patients in their testing and training sets than the Asian countries due to a combination of fewer patients enrolled at those sites and proportionately fewer patients with known etiologies. In leave-one-site-out validation testing, we found that the climate test tends to outperform the seasonality test, and that there were notable differences in c-statistics between sites with the order of performance similar to within rolling-origin-recalibration evaluation (*Figure 4—figure supplement 2*).

### Addition of a diagnostic test to integrative models improves discrimination

Emerging efforts to develop diagnostic devices, including laboratory assays as well POC tests, have focused on the performance of the test used in isolation. Here, we consider the use of a diagnostic device in combination with clinical predictive models. We used the integrative model to examine the impact that an additional diagnostic would have on discrimination of two of the best performing models. We show that an additional diagnostic, with varying sensitivity and specificity, would improve the cross-validated AUC as expected (*Table 3*). An additional test with a 70% sensitivity and 70% specificity increases the AUC by 3–5%, while a more specific test could increase the AUC by 10%.

We next examined ROC curves, which visually demonstrate the effect of additional diagnostics with varying levels of sensitivity and specificity (*Figure 5*). We show that a similar level of sensitivity and specificity is achievable by the model with the pre-test information versus the model with seasonal information. Additionally, the additional diagnostics result in improved overall sensitivity and specificity corresponding to sensitivity and specificity of the diagnostic. The overall sensitivity and specificity of each model is greater than the diagnostic alone.

**Table 3.** AUC and 95% confidence intervals from 80% training set after adding an additional point-of-care diagnostic test with specified sensitivities (Se.) and specificities (Sp.) to the current patient test and pre-test odds.
Additionally, + and - refer to our model indicating a true positive or false positive, respectively, based on the threshold for each model which achieves a 0.90 or 0.95 specificity. Only patients who were prescribed/given antibiotics are included in the count.

| Model | Addl. diag. (Se.,Sp.) | Auc (95% CI) | Specificity=0.90 | | Specificity=0.95 | |
|---|---|---|---|---|---|---|
| | | | True + | False + | True + | False + |
| Pre-test * PresPtnt | None | 0.839 (0.809–0.869) | 88 | 29 | 60 | 16 |
| | (0.7, 0.7) | 0.876 (0.849–0.902) | 102 | 31 | 78 | 16 |
| | (0.7, 0.95) | 0.933 (0.914–0.952) | 132 | 31 | 123 | 16 |
| | (0.9, 0.95) | 0.972 (0.960–0.984) | 154 | 34 | 147 | 18 |
| PresPtnt * Seasonal | None | 0.830 (0.798–0.861) | 70 | 25 | 54 | 11 |
| | (0.7, 0.7) | 0.870 (0.842–0.897) | 101 | 27 | 68 | 14 |
| | (0.7, 0.95) | 0.931 (0.912–0.951) | 130 | 27 | 121 | 16 |
| | (0.9, 0.95) | 0.971 (0.959–0.984) | 154 | 30 | 149 | 18 |
| PresPtnt | None | 0.809 (0.776–0.842) | 66 | 31 | 41 | 15 |
| | (0.7, 0.7) | 0.857 (0.827–0.886) | 98 | 33 | 68 | 16 |
| | (0.7, 0.95) | 0.925 (0.904–0.946) | 129 | 33 | 117 | 18 |
| | (0.9, 0.95) | 0.968 (0.955–0.981) | 153 | 34 | 149 | 18 |

The online version of this article includes the following source data for Table 3:
**Source data 1.** Frequency table of pathogens in which the post-test odds formulation with varying specifity (Sp.) chosen have false positives.

## Breaking the conditional independence assumption can be addressed using 2-D Kernel density estimates

Our integrative post-test odds method assumes the conditional independence of its component tests, and thus we performed simulation of increasingly conditionally dependent components to assess the performance of the method when the assumption is broken. We showed that the AUC of the post-test odds model deteriorates quickly as the conditional independence assumption is violated (*Table 4*). With no conditional dependence between predictions from models X and Y, the result using one-dimensional kernel density is comparable to the result with two-dimensional kernel density model. However, as the conditional correlation between the tests increase to −0.90, the one-dimensional AUC decreases by about 11% while the post-test odds with the two-dimensional test performs consistently across this range of conditional correlation.

## Clinician use of an integrative predictive model for diarrhea etiology could result in large reductions in inappropriate antibiotic prescriptions

Given that one potential application of an integrative predictive model for diarrhea etiology would be as support for clinical decision making for antibiotic use (i.e. antibiotic stewardship), we then examined the impact that the top predictive model would have on prescription of antibiotics by clinicians in GEMS. Of the 3366 patients included in our study, 2653 (79%) were treated with or prescribed antibiotics, 806 (30%) of whom were prescribed to those with a viral-only etiology as determined by qPCR. Here, we examined how use of integrative predictive model could have altered antibiotic use in our sample. Of the 681 patients in the 20% test set, 540 (79%) were prescribed antibiotics, including 166 (30%) with a viral-only etiology. Of those prescribed/given antibiotics the model with pre-test odds, with threshold chosen for an overall specificity of 0.90, identified 88 (53%) viral cases as viral, and 29 non-viral cases as viral. With an additional diagnostic with a sensitivity and specificity of 0.70, the same model would on average identify 102 (61%) viral cases as viral with the same 31 non-viral cases identified as viral. Assuming that clinicians would not prescribe antibiotics for those cases identified by the predictive model with the additional diagnostic as viral, we would avoid 88 (53%) and 102 (61%) of inappropriate antibiotic prescriptions in the two scenarios described. The majority of the false positives (29 and 30 in the two scenarios) were episodes majority attributed to Shigella, ST-ETEC, and combinations of rotavirus with a non-viral pathogen (*Table 3—*

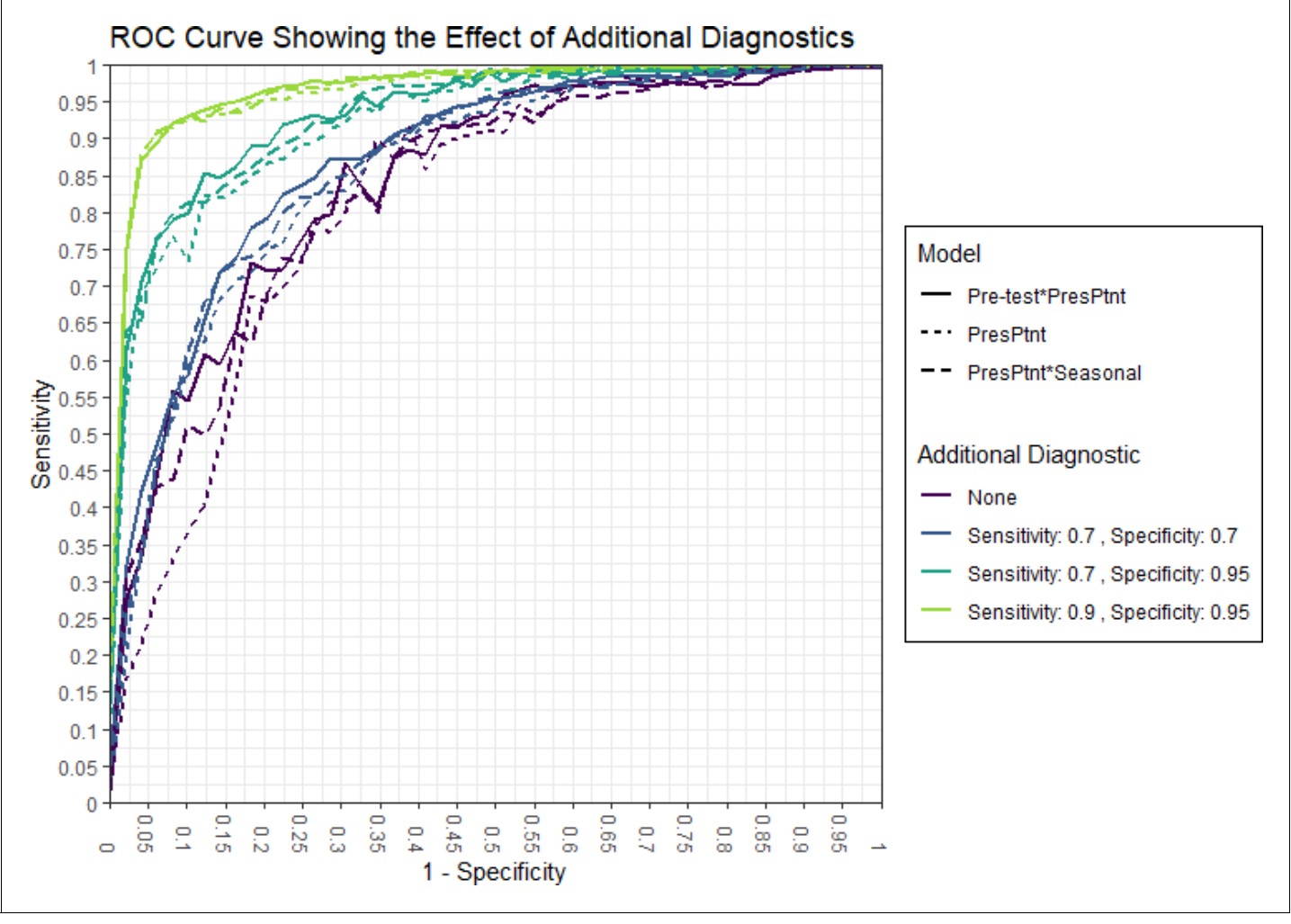

**Figure 5.** ROC curves from validation from 80% training set. Curves shown for three models with additional diagnostics.

**Table 4.** Average AUC's from one-dimensional and two-dimensional kernel density estimates (KDE) when the post-test odds conditional independence assumption is broken.

The table shows the factor ($\gamma$) used to simulate induced conditional dependence between two covariates and their average conditional correlation. Additionally, it shows the average AUC resulting from a post-test odds model where a one-dimensional kernel density estimate (conditional independence assumed) is generated for each covariate, and a post-test odds model where a two-dimensional joint kernel density estimate is derived for the two covariates.

| | | AUC | |
|---|---|---|---|
| $\gamma$ | $cor(X, Y \| Z)$ | **1D-KDE** | **2D-KDE** |
| -2.000 | −0.894 | 0.725 | 0.830 |
| -1.000 | −0.709 | 0.758 | 0.828 |
| -0.500 | −0.446 | 0.824 | 0.838 |
| 0.000 | 0.002 | 0.838 | 0.836 |
| 0.500 | 0.448 | 0.836 | 0.836 |
| 1.000 | 0.708 | 0.831 | 0.840 |
| 2.000 | 0.894 | 0.810 | 0.836 |

*source data 1*). All of these false positive, with exception of 1 case, had non-bloody diarrhea, and thus would have been deemed as not requiring antibiotics by WHO IMCI guidelines.

## Discussion

The management of illness in much of the world relies on clinical decisions made in the absence of laboratory diagnostics. Such empirical decision-making, including decisions to use antibiotics, are informed by variable degrees of clinical and demographic data gathered by the clinician. Traditional clinical prediction rules focus on the clinical data from the presenting patient alone. In this analysis, we present a method that allows flexible integration of multiple data sources, including climate data and clinical or historical information from prior patients, resulting in improved predictive performance over traditional predictive models utilizing a single source of data. Using this formulation, if certain sources of data such as climate or previous patient information are not available (e.g. due to a lack of internet connection or data infrastructure), the prediction can still be made using current patient information or seasonality, as appropriatethe other sources. A mobile phone application is an ideal platform for a decision support tool implemented in low-resource settings. Through internet access by wifi or cellular data, a smartphone platform could automatically download recent patient or climate data, while its portability would facilitate clinicians in entering current patient clinical information. We show that application of such a predictive model, especially with an additional diagnostic test, may translate to reductions in inappropriate antibiotic prescriptions for pediatric viral diarrhea.

The global burden of acute infectious diarrhea is highest in low- and middle-income countries (LMICs) in southeast Asia and Africa (*Walker et al., 2013*), where there is limited access to diagnostic testing. The care of children in these regions could greatly benefit from an accurate and flexible decision making tool. Decisions for treatment are often empiric and antibiotics are over-prescribed (*Rogawski et al., 2017*), although the majority of cases of diarrhea do not benefit from antibiotic use and also many instances of acute watery diarrhea are self-limiting . For example, 2653 (79%) of the 3366 patients in our study were treated with or prescribed antibiotics. Of these 806 (30%) were prescribed to those with a viral-only etiology. Unnecessary antibiotic use exposes children to significant adverse events including serious allergic reactions (*Logan et al., 2016*, *Marra et al., 2009*) and clostridium difficile infection (*Jernberg et al., 2010*), and contributes to increased antimicrobial resistance. We show that a predictive model can be used to discriminate between those with and without a viral-only etiology and that the inappropriate use of antibiotics can be avoided in 54% cases using our model with no additional diagnostics.

We found using within rolling-origin-recalibration evaluation that models which include either the pre-test odds calculated historical rates or the seasonal test were the best at discriminating between viral etiologies and other etiologies, a finding that held true across training and testing set sizes. However, in the leave-one-out validation, models which included the alternate pre-test odds and climate tended to perform the best. This difference is likely due to the generalizeability of the individual tests, i.e, the leave-one-out tests are trained at the continental level and the effect of climate on etiology is intuitively more generalizeable than seasonal curves which are very specific to each location. We found that our integrative model with only the historical (pre-test) information included (without additional diagnostics) would have identified a viral-only etiology in 88 (53%) patients who received antibiotics. We then show that even the use of an additional diagnostic test with modest performance (70% sensitivity and specificity) would further decrease inappropriate antibiotic use by another 14 (for a total of 102, or 61% of) patients. In the context of calls by the WHO for the development of affordable rapid diagnostic tools (RDTs) for antibiotic stewardship (*Declaration, 2017*), our findings suggest that development and evaluation of novel RDTs should not be performed in isolation. Potential for integration of rapid diagnostic tests into clinical prediction algorithms should be considered, although this needs to be balanced with the additional time and resources needed. The incremental improvement in discriminative performance achieved by the addition of an RDT to a clinical prediction algorithm may not be cost-effective in lower resourced settings. Finally, providing this model in the form of a decision support tool to the clinician could translate to reductions in inappropriate use of antibiotics, although further research needs to be done to explore the degrees of certainty that clinicians require to alter treatment decisions.

The novel use of kernel density estimates to derive the conditional tests when calculating the post-test odds enabled a flexibility in model input. While kernel density estimates have been used for conditional feature distributions in Naïve Bayes classifiers (*John and Langley, 1995*, *Murakami and Mizuguchi, 2010*), here we show that they can be used to derive conditional likelihoods for diagnostic tests constituting one or more features, stressing the effect of the overall test on the post-test odds and not individual features. As such, complicated machine learning models can be combined with simple diagnostics as part of the post-test odds. For example, we could have fit neural networks in lieu of logistic regression models, and in addition to these more complicated models, it is possible to incorporate the result of an RDT that make results available to the clinician at the point-of-care. Additionally, our method of using two-dimensional kernel density estimates can also be used to overcome the conditional independence assumption for tests based on potentially interrelated diagnostic information. Densities with higher than two dimensions can be considered, though, computational limitations are likely in both speed and, we expect, accuracy, as the dimensions increase.

Our study has a number of limitations. First, a robust training set of both cases and non-cases is required to adequately build the conditional kernel densities. Second, the post-test odds calculation, at the time of prediction, lacks interpretation on a feature level like a logistic regression or decision tree. Although, we do observe the effect of a test on an observation, we cannot see which features caused that effect without diving deeper into the training of the diagnostic tests. Thirdly, the prediction algorithm generated by the post-test odds model using GEMS data was only validated internally, and further studies are need for external validation and field implementation. Fourth, our estimation of antibiotic use reduction used data from a clinical research study, which may have biases inherent to such studies. Last, our study uses the AFe cut-off of greater than or equal to 0.5 to assign etiology from the qPCR data. This cutoff was selected based on expert elicitation, but the effect of using this cut-off has not been explored. Bacterial cases with AFe¡0.5 were excluded in our analysis, but may still benefit from antibiotic treatment.

In conclusion, we have developed a clinical prediction model that integrates multiple sources external to the presenting patient, through use of a post-test odds framework and showed that it improved diagnostic performance. When applied to the etiological diagnosis of pediatric diarrhea, we demonstrate its potential for reducing inappropriate antibiotic use. The flexible inclusion or exclusion of output from its components makes it ideal for decision support in lower resourced settings, when only certain data may be available due to limitations in information computation or connectivity. Additionally, the ability to incorporate new training data in real-time to update decisions allows the model to improve as more data is collected. Such a predictive model has the potential to improve the management of pediatric diarrhea, including the rational use of antibiotics in lower resourced settings.

## Acknowledgements

This investigation was supported by the University of Utah Study Design and Biostatistics Center, with funding in part from the National Center for Research Resources and the National Center for Advancing Translational Sciences, National Institutes of Health, through Grant 8UL1TR000105 (to BJB, BH, and TG). Research reported in this publication was supported by the NIAID of the NIH under award number R01AI135114 (to DTL), and the Bill and Melinda Gates Foundation award OPP1198876 (to DTL). The authors would like to thank Bill and Melinda Gates for their active support (JLP and DC) of the Institute for Disease Modeling and their sponsorship through the Global Good Fund.

## Additional information

### Funding

| Funder | Grant reference number | Author |
| --- | --- | --- |
| National Center for Advancing Translational Sciences | 8UL1TR000105 | Ben J Brintz<br>Benjamin Haaland<br>Tom Greene |

| National Institute of Allergy and Infectious Diseases | R01AI135114 | Daniel T Leung |
| Bill and Melinda Gates Foundation | OPP1198876 | Daniel T Leung |

The funders had no role in study design, data collection and interpretation, or the decision to submit the work for publication.

### Author contributions

Ben J Brintz, Conceptualization, Data curation, Formal analysis, Validation, Investigation, Visualization, Methodology, Writing - original draft, Writing - review and editing; Benjamin Haaland, Conceptualization, Visualization, Methodology, Writing - review and editing; Joel Howard, Conceptualization, Writing - review and editing; Dennis L Chao, Joshua L Proctor, Tom Greene, Methodology, Writing - review and editing; Ashraful I Khan, Sharia M Ahmed, Lindsay T Keegan, Adama Mamby Keita, Eric J Nelson, Adam C Levine, Andrew T Pavia, Writing - review and editing; Karen L Kotloff, Resources, Writing - review and editing; James A Platts-Mills, Data curation, Writing - review and editing; Daniel T Leung, Conceptualization, Resources, Supervision, Funding acquisition, Writing - review and editing

### Author ORCIDs

Ben J Brintz (ID) https://orcid.org/0000-0003-4695-0290
Daniel T Leung (ID) https://orcid.org/0000-0001-8401-0801

### Decision letter and Author response

Decision letter https://doi.org/10.7554/eLife.63009.sa1
Author response https://doi.org/10.7554/eLife.63009.sa2

## Additional files

### Supplementary files

- Transparent reporting form

### Data availability

GEMS data are available to the public from https://clinepidb.org/ce/app/. Data and code needed to reproduce all parts of this analysis are available from the corresponding author's GitHub page: https://github.com/LeungLab/GEMS_PostTestOdds [copy archived at https://archive.softwareheritage.org/swh:1:rev:67f4f5a0cdc3d569e756142f0142aaa23a9b1e03/].

The following previously published dataset was used:

| Author(s) | Year | Dataset title | Dataset URL | Database and Identifier |
|---|---|---|---|---|
| Gates Enterics Project, Levine MM, Kotloff K, Nataro J, Khan AZA, Saha D, Adegbola FR, Sow S, Alonso P, Breiman R, Sur D, Faruque A | 2018 | Study GEMS1 Case Control | https://clinepidb.org/ce/app/record/dataset/DS_841a9f5259#Contacts | ClinEpiDB, DS_841a9f5259 |

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

## Appendix 1

### Weighted weather station data

Daily local weather information was constructed based on data from weather stations within 200 km of the site of interest. We chose 200 km because one our sites, Mozambique, does not have any stations nearer than 180 km. We then collect the temperature and rain info from the top five closest weather stations and take a weighted average where they are weighted inversely by distance so that the closer weather stations will have more effect on the average. For instance, for temperature on day $d$ across the five closest weather stations: $T_{d.} = \frac{\sum_{i=1}^{5} T_{di} \cdot d_i^{-1}}{\sum_{i=1}^{5} d_i^{-1}}$ where $T_{di}$ is the average temperature for weather station $i$ on day $d$ and $d_i$ is the distance from weather station $i$.

## Appendix 2

### Pre-test odds from prior patient predictions for prediction in new sites

We calculated pre-test odds by combining past predictions from predictive model A, the presenting patient model. By taking a weighted average of the recently predicted odds of viral etiology, we attempt to capture recent local trends in diarrhea pathogens, such as localized outbreaks. This is similar to heuristic decision making historically used by clinicians. We aggregated the odds calculated from the presenting patient model on their probability scale for each site over the past d days such that pre-test probability $\pi_d$ for day $d$ is

$$
\begin{aligned}
\pi_d &= \frac{P_{d-n+1} \cdot w_1 + P_{d-n+2} \cdot w_2 + \cdots + P_d \cdot w_n}{w_1 + w_2 + \cdots + w_n} \\
P_d &= \frac{1}{k} \Sigma_{i=1}^{k} P_{di}
\end{aligned}
$$

where $P_{di}$ are the $i = 1, \cdots, k$ current patient predictions converted from the odds scale to the probability scale on day $d$ and $n$ is the number of prior days included in the calculation. Provided the greatest weights are put on the most recent predictions, we would expect an influx of certain symptoms related to a viral etiology to be represented by $\pi_d$.

