## [Decision Letter]

**Acceptance summary:**

We were excited by your idea to integrate population level data such as seasonality and recent weather into clinical decision making tools. This approach has potential widespread utility for multiple pathogens with seasonal variability, and even for the SARS-CoV-2 pandemic which is notable for wide spatiotemporal incidence of new infections. In the case of pediatric diarrhea, the potential to limit antibiotic overprescribing is of enormous public health importance.

**Decision letter after peer review:**

Thank you for submitting your article "A modular approach to integrating multiple data sources into real-time clinical prediction for pediatric diarrhea" for consideration by *eLife*. Your article has been reviewed by two peer reviewers, and the evaluation has been overseen by a Reviewing Editor and a Senior Editor. The following individual involved in review of your submission has agreed to reveal their identity: Joe Brown (Reviewer #1).

The reviewers have discussed the reviews with one another and the Reviewing Editor has drafted this decision to help you prepare a revised submission.

Summary:

This paper describes a computationally advanced method for clinical prediction of pediatric diarrhea cases based on the inclusion of prior data on the etiology of disease and additional data beyond that available at patient presentation. Over prescription of antibiotics is a global issue and major contributor to antimicrobial resistance. This paper's methods would allow more accurate diagnosis of viral causes of diarrhea based not just on individual data but also on pre-existing distributions of cases in other patients and on regional factors such as seasonality.

The approach is sound and the conclusions justified. The authors present a flexible approach to predictive modeling of viral diarrhea that is globally adaptable to local conditions and available resources. The work is extensive and thorough with major thought given to sensitivity of model validation and calibration. Though the difference in AUC with these added non-patient specific variables is only slight, inclusion into bedside diagnostic tools could decrease the unnecessary use of antibiotics significantly.

The statistical methodology is rigorous and extensive with substantial detail given to model assumptions and validation

Essential revisions:

1) The utility of the method depends on the quality (and diagnostic accuracy) of prior data that may not generally be available in high-burden settings. Models rely on highly specific quantitative pathogen data that may not be widely available outside of the dataset that the authors use. The GEMS quantitative re-analysis from which data were derived (Liu et al., 2016) argues convincingly that quantitative data are required to establish etiology of cases, given the high prevalence of asymptomatic carriage of enteric pathogens in children in high-burden cohorts. This does not necessarily preclude the utility of this analysis, however, given that more such data may become available in high-burden settings. But, it should be acknowledged that the current approach as described is likely to be restricted to settings where high quality, accurate prior etiological data are available. Currently that excludes most high-burden sites.

A sensitivity analysis that accounts for sub-optimal prior etiology data on study outcomes would be a short but useful addition to the current analysis. That is: how does the model perform when prior / training data are less than 100% accurate?

2) A reader may wonder whether a regional or global dataset on prior etiology could substitute for local data in training models for clinical prediction, given the current and near-term scarcity of available local data at specific sites of interest. A straightforward way to test this would be to train models in country A, same region (e.g., Bangladesh) and test the predictive model against the cohort from country B, same region (e.g., Pakistan). Such an analysis would help convey to readers how local the training data need be to improve diagnostic accuracy / reduce prescription of unnecessary antibiotics. Similarly, a model trained using data taken from all GEMS sites could be applied to specific countries, in order to assess model performance when no reliable, locally available data on etiology exist. That may well be the norm in many settings of interest.

3) Of 9439 children, only 3366 had known etiologies which indicates the likely true cause of diarrhea remains indeterminate for almost 2/3 of cases. Unknown etiologies were not included in models (AFe>=0.5 only), so authors don't consider diarrheal episodes with no clear majority but where say AFe viral=0.49 and AFe bacterial=0.48. Such cases could be important to evaluate particularly if they are highly prevalent and they may still benefit from antibiotic treatment. Please consider describing this as a limitation or performing sensitivity analyses go account for the uncertainty of the AFE cutoff.

---

## [Author Response]

Essential revisions:1) The utility of the method depends on the quality (and diagnostic accuracy) of prior data that may not generally be available in high-burden settings. Models rely on highly specific quantitative pathogen data that may not be widely available outside of the dataset that the authors use. The GEMS quantitative re-analysis from which data were derived (Liu et al., 2016) argues convincingly that quantitative data are required to establish etiology of cases, given the high prevalence of asymptomatic carriage of enteric pathogens in children in high-burden cohorts. This does not necessarily preclude the utility of this analysis, however, given that more such data may become available in high-burden settings. But, it should be acknowledged that the current approach as described is likely to be restricted to settings where high quality, accurate prior etiological data are available. Currently that excludes most high-burden sites.A sensitivity analysis that accounts for sub-optimal prior etiology data on study outcomes would be a short but useful addition to the current analysis. That is: how does the model perform when prior / training data are less than 100% accurate?

Thank you for suggesting this additional analysis. We have included a sensitivity analysis in which we use only “conventional diagnostic methods” data for the pre-test odds calculation, instead of the qPCR data. The conventional methods, as described in the initial GEMS report (Kotloff et al., 2013), while more likely to be available in high-burden settings, identified fewer pathogens as attributable causes of diarrhea, and are regarded to be less accurate than qPCR with regards to pathogen attribution. We additionally provide alternatives to using prior etiology data such as the climate component that can be trained at the regional level with results shown in the supplement.

We added “Additionally, we include a sensitivity analysis by calculating the pre-test odds using conventional diagnostic methods data, as qPCR data are unlikely to be available in high-burden settings” to the Materials and methods section and “Using conventional diagnostic methods data data to calculate pre-test odds instead of qPCR data reduces AUC slightly from.839 to 0.829 (0.798 0.860).” to the Results section.

2) A reader may wonder whether a regional or global dataset on prior etiology could substitute for local data in training models for clinical prediction, given the current and near-term scarcity of available local data at specific sites of interest. A straightforward way to test this would be to train models in country A, same region (e.g., Bangladesh) and test the predictive model against the cohort from country B, same region (e.g., Pakistan). Such an analysis would help convey to readers how local the training data need be to improve diagnostic accuracy / reduce prescription of unnecessary antibiotics. Similarly, a model trained using data taken from all GEMS sites could be applied to specific countries, in order to assess model performance when no reliable, locally available data on etiology exist. That may well be the norm in many settings of interest.

Thank you for highlighting this – our submission included a “leave-one-out cross-validation” in Figure 4—figure supplement 2 in which we have trained at the continent (“region”) level and tested on the left-out site. We describe this in the Materials and methods subsection “Validating the method”. These findings are included in the text in the Results section, “In leave-one-site-out cross-validation testing, we found that the climate test tends to outperform the seasonality test”. Additionally, Table 2 contains country-specific results for the present patient model alone, which is trained on 80% of each country (though the country of interest is included in that training).

3) Of 9439 children, only 3366 had known etiologies which indicates the likely true cause of diarrhea remains indeterminate for almost 2/3 of cases. Unknown etiologies were not included in models (AFe>=0.5 only), so authors don't consider diarrheal episodes with no clear majority but where say AFe viral=0.49 and AFe bacterial=0.48. Such cases could be important to evaluate particularly if they are highly prevalent and they may still benefit from antibiotic treatment. Please consider describing this as a limitation or performing sensitivity analyses go account for the uncertainty of the AFE cutoff.

We agree that more exploration of AFe cut-off is a limitation of our study. We have now added into the limitations paragraph of the Discussion:

“Last, our study uses the AFe cut-off of greater than or equal to 0.5 to assign etiology from the qPCR data. This cutoff was selected based on expert elicitation, but the effect of using this cut-off has not been explored. Bacterial cases with AFe<0.5 were excluded in our analysis, but may still benefit from antibiotic treatment.”